# Eating Healthy: Understanding Added Sugar through Proportional Reasoning

**DOI:** 10.3390/ijerph182312821

**Published:** 2021-12-05

**Authors:** Debasmita Basu, Hong B. Nguyen

**Affiliations:** 1Department of Natural Sciences and Mathematics, Eugene Lang College of Liberal Arts, The New School, New York, NY 10011, USA; 2Department of Psychology, The New School for Social Research, The New School, New York, NY 10003, USA; nguyh376@newschool.edu

**Keywords:** proportional reasoning, public health, food nutrition, added sugar, teaching and learning of mathematics and science, STEM education, integrated curriculum

## Abstract

Research suggests that integrated STEM activities can best support students in developing their mathematical and scientific understanding. On one hand, while science provides mathematics with real-life authentic problems to investigate, mathematics provides science powerful tools to explore those problems. In line with this call, in this study, we designed an integrated lesson at the cross-section of proportional reasoning and added sugar present in food products to explore how added sugar provides students with a meaningful context to engage in proportional reasoning and how proportional reasoning helps students identify the quantity of added sugar present in different food products and provides students with a platform to initiate a conversation around quality of food products. Developed on the theoretical framework of Realistic Mathematics Education (RME), this lesson was remotely implemented on three middle school students. The result section highlights the design principle of the lesson that provided students with an opportunity to construct an understanding of both the disciplines through a mutual interaction.

## 1. Introduction

Over the years, an increased emphasis has been given to the integration of mathematics and science curricula. According to NCTM [1] and NRC [2], a well-coordinated science and mathematics curriculum can provide learners with a meaningful learning experience and develop an in-depth understanding of both the disciplines. A similar argument was also made by Rutherford and Ahlgren [3], who said, “Science provides mathematics with interesting problems to investigate, and mathematics provides science with powerful tools to use in analyzing them” (pp. 16–17). Further, advocates of integrated education argue that in real life, situations do not involve only a single discipline; hence, an integrated science and mathematics curriculum will reduce students’ fragmented learning experience and help them to recognize the relevance of different scientific and mathematical concepts in their lives [4,5]. Keeping in mind the importance of integrated education and its implication on students’ out-of-school lives, in this study, we developed a lesson at the cross-section of the health science of added sugar and the mathematical concept of proportional reasoning and explored,
Does a task developed on added sugar provide students with a meaningful context to engage in proportional reasoning?Does proportional reasoning help students identify the quantity of added sugar present in different food products and thus engage students in a discussion about food quality?

## 2. Literature Review

### 2.1. M of STEM: Proportional Reasoning

Proportional reasoning entails a multiplicative relationship between two quantities. Two quantities are proportional if the ratio between the quantities remains the same irrespective of their actual measures [6]. For example, lemon syrup and water in two different brands of lemonade (lemonade A and lemonade B) will be proportional if lemon syrup and water maintain the same ratio in the two lemonades (lemonade A: lemon syrup 5 oz, water 15 oz, the ratio between lemon syrup and water = 5:15 = 1:3; lemonade B: lemon syrup 10 oz, water 30 oz, the ratio between lemon syrup and water = 10:30 = 1:3). Proportional reasoning is considered one of the essential topics in middle school mathematics, laying the ground for understanding many higher-level concepts in mathematics, science, economics, and geography [7,8,9]. Furthermore, the concept of proportionality is important in professions such as architecture, nursing, and pharmacy [10]. Although proportional reasoning is a fundamental concept foundational to the understanding of higher mathematical and scientific concepts, reasoning between proportional quantities is not intuitive to students. Students often struggle to distinguish between proportional and non-proportional situations [6,11] and, as a result, fail to reason multiplicatively and use additive thinking in situations involving proportional quantities [12,13]. Researchers attributed such lack of development of students’ proportional concept to the ways proportional reasoning is taught in schools. According to Sowder et al. [12], teachers with strong mathematical background often struggle to distinguish between additive and multiplicative situations and thus fail to reconceptualize their knowledge of proportional reasoning. Furthermore, in schools, teaching of mathematical concepts in isolation does not prepare the learners to connect topics and learn mathematics meaningfully [10,14]. Discussing the effectiveness of different curricula in developing students’ proportional reasoning, Ben-Chaim, Fey, Fitzgerald, Benedetto, and Miller [15] mentioned that context plays a major role in students’ performance in proportional reasoning tasks. They said that familiar and authentic contexts are capable of engaging students in reasoning around proportional quantities and “developing their own repertoire of sense-making tools to help them to produce creative solutions and explanations” [15] (p. 271). Consistent with these authors, Misnasanti, Utami, and Suwanto [16] said that there exists strong evidence that students’ engagement in proportional reasoning increases when mathematical problems are presented in authentic real-life contexts. With a similar goal in mind, in this study, we provided students with a familiar problem setting of added sugar in food products and explored the ways added sugar provides students with a meaningful context to engage in proportional reasoning.

### 2.2. S of STEM: Added Sugar

Research shows that the food products featured during children’s popular programs are usually loaded with added sugar, which influences children’s food purchase requests [17,18,19]. Adults are often unaware of the nutrient contents of food and their associated health implications, and as a result, they frequently end up buying food products that are unhealthy for children [20]. Such food products, rich in added sugar and other unhealthy components, have significantly contributed to the increasing rate of obesity and other cardiovascular diseases amongst children [21,22,23]. According to the Center for Disease Control (CDC), the percentage of children and adolescents between ages 2 and 19 years affected by obesity has more than tripled since the 1970s. In 1971–1974, while the percentage of children and adolescents identified with the condition of obesity was 5.2%, in 2015–2016, the percentage increased to 18.5% [24]. To address the growing issue, health professionals and researchers have advised an early intervention; they emphasized the inclusion of food and nutrition education in the early years of schooling [25]. Suggestions have also been made to use food nutrition as a tool to teach other disciplines [26].

Considering the severity of the issue, researchers, such as James and Adams [27] and Hyman [28], advocated a partnership between public health and mathematics education. They said an integrated curriculum would provide learners with a meaningful context to learn mathematical concepts, develop students’ quantitative skills, and instill a sense of responsibility towards making healthy life choices. However, such a curriculum, developed at the cross-section of mathematics and food nutrition, is limited. Hovland et al. [26] developed the Food, Math, and Science Teaching Enhancement Resource (FoodMASTER) project, where they developed a 10-chapter curriculum to integrate food and health to teach different science and mathematics concepts.

The Food Fact lesson has been developed to engage students in discussion around the quantity of added sugar in food products and prompts students to use proportional reasoning to evaluate if we are consuming excessive amounts of added sugar through food, thus risking developing obesity and other health conditions. We hoped that the lesson would help the learners initiate a conversation around added sugar and its health implications. Based on the aim of the study stated above, this research utilized the theoretical framework of Realistic Mathematics Education (RME) to explore how the real-life context of added sugar created a natural platform to engage students in a meaningful mathematical experience and use mathematics as a potential tool to develop an in-depth understanding of the health-science issue. In the following paragraph, first, we discuss how RME has been described by researchers and has informed our study, followed by a brief description of the lesson.

## 3. Theoretical Framework

In this study, we used the theoretical framework of Realistic Mathematics Education (RME). Realistic Mathematics Education (RME) is a “domain-specific instruction theory for mathematics” [29] (p. 521). Opposing the traditional, mechanistic approach to mathematics, in the 1960s, Freudenthal proposed Realistic Mathematics Education, which advocates linking mathematics learning to real-world contexts. Instead of introducing a mathematical concept through its symbolic representations and theoretical abstractions, RME starts by identifying the problems students must deal with in their daily lives and their knowledge. In the process, the role of mathematics teachers is to translate the abstract mathematical objects to concrete situations and help the students to solve those situations by asking them to identify and develop effective solutions [30,31,32]. Teachers need to reconsider their beliefs about the role of out-of-school knowledge and everyday skills in solving mathematical problems and see mathematics incorporated in the real world as a gateway to formal mathematics learning [33].

According to Verschaffel, Greer, and De Corte [34], students often nurture a resilient attitude towards school mathematics and its expectations and exhibit a tendency to segregate formal mathematics from their everyday skills and knowledge. Such notion refrains students from sense making of formal mathematics in typical school settings. Defying the nature of traditional mathematics, RME suggests that students are more likely to learn mathematics if they can actively process information and construct knowledge through active exploration of authentic problem situations [35]. Such practices prepare students to communicate mathematically and make connections with each other and the real world. This type of process-oriented mathematics should ensure that students are able to apply what they have learned in one situation to another situation, which is a key goal in mathematics education [36,37].

Researchers have identified six core principles of teaching mathematics in connection to RME:The activity principle advocates that mathematics can be best learned by doing. Freudenthal [35,38] said that ready-made mathematics, through an anti-didactic intervention, cannot be transferred to students. Rather, mathematics can be best learned through an active participation [39]. It provides students with an opportunity to come up with their own strategies and insights to address a real-life situation.In addition to students taking agency of their own learning, the second principle of RME emphasizes connecting mathematics to reality. As the reality principle suggests, mathematical learning should not commence from numerical strategies and formulae followed by their applications in real life; rather, our reality should serve the learners with sources to learn mathematics. Instead of beginning with abstractions, Freudenthal [35,38] proposed “didactical phenomenology” in which “one is concerned with describing how a mathematical idea emerges in a learning and teaching process as a means to organize phenomena” [40] (p. 25).The level principle refers to the various levels of mathematical understanding and cognitive development that students obtain as they engage with some real-world scenarios to formal mathematical setups. According to this principle, RME bridges the students’ informal experience with pure mathematical knowledge and allows students to construct their own instruction and use their own informal strategies to learn formal math. Thus, students will not only be able to develop their own individual learning path but also mature through the process.The intertwinement principle suggests mathematical content domains, such as number, geometry, data, space and measure, algebra, and probability, should be taught together to the extent possible. This enables learners to make comparisons among mathematical tools and to see relationships between topics.The interactivity principle indicates that learning mathematics is a personal activity and one that involves interaction with other people. It is the interaction with other people that helps to stimulate and develop a learner’s mathematical abilities. This principle of Realistic Mathematics Education allows for the understanding that mathematics cannot be learned alone; instead, it must develop through social interactions.The guidance principle implies that RME teachers should play a more active role in their students’ learning. Mathematical programs should contain scenarios that can help students develop long-term learning trajectories.

Out of these six principles of RME, this study is guided by the first two principles: the activity principle and the reality principle.

Consistent with the reality principle, the study has been motivated by the growing condition of obesity and children’s awareness around it. Added sugar has a significant contribution to obesity and other health conditions, and to make students aware of the impacts of added sugar we designed three tasks in which we hoped that the rich context would serve the students with an authentic investigative opportunity to develop their own mathematical reasoning. To gain students’ interest, the study began by showing food products that are a part of their daily lives, such as cookies and yogurt. Afterward, the researcher asked some general questions about the students’ relationship with these products, which then led to the introduction of food nutrition, especially added sugar. The students then solved the tasks in which they used proportional reasoning to calculate and compare added sugar present in different food products and decided their nutritional values. As the reality principle suggests, the tasks did not commence with any mathematical abstraction. Rather, the context of added sugar significantly assisted the students to learn mathematics meaningfully.

In this study, when we designed the three tasks, one of our goals was to actively engage students in proportional reasoning to explore the nutritional values of the food products and decide for themselves if the products were healthy. Consistent with the activity principle, we hoped that as students calculate the quantity of added sugar, they would identify the proportional relationships that exist between the different quantities and use the particular concept to solve the problems. We did not intend to prompt students to use any particular mathematical concept and decided to let students come up with their own strategies as they engage with the tasks. We hoped that the tasks would improve students’ ability to solve math problems by engaging them in critical learning activities that enhance their critical thinking skills.

### Task Design and Analytical Framework-Proportional Reasoning

Several studies have identified the different ways to assess students’ proportional reasoning. For example, Lamon [6] identified four semantic problem types typically organized by proportion. The four problem types are:Well-chunked measures: Comparison of two extensive resulting in an intensive measure (rate). Example: Speed equals miles per hour.Part-Part-whole: The extensive measure of the single subset of a whole is given in terms of the cardinalities of two sub-subsets of which it is composed. Example: Ratio of carbohydrate, protein, and fat in a food product.Associated sets: Sometimes, the relationship between two elements is unknown unless their relationship is defined within the problem situation. For example, the connection between teaspoons and sugar/servings is not apparent unless a statement defining the relation between the two quantities is explicitly made.Stretchers and Shrinkers: When a continuous one-to-one ratio preserving mapping exists between two quantities, then the situation containing the quantities involves scaling up (Stretchers) or scaling down (Shrinkers).

Cramer and Post [11] also classified proportional reasoning tasks into three broad categories to assess students’ reasoning around proportional quantities. The three categories are: (1) missing values problems, (2) numerical comparison problems, and (3) qualitative prediction and comparison problems. In missing value problems, students are given three pieces of information, and they are asked to find the fourth value, while in numerical comparison problems, students are given two rates, and they are asked to compare the rates. The third type, qualitative prediction and comparison problems, relies on students’ qualitative reasoning skills to engage in proportional reasoning, which, according to the authors, helps students understand the meaning of proportion and helps them improve their problem-solving ability.

This study utilized parts of the above two frameworks by Lamon [6] and Cramer and Post [11] to design a set of proportional reasoning tasks utilizing the rich context of added sugar. The tasks are consistent with the associated sets tasks and stretchers and shrinkers as defined by Lamon [6] and missing values problems as discussed by Cramer and Post [11]. In accordance with Lamon and Cramer and Post, we determined the following problem types would help us recognize students’ reasoning between proportional quantities. In the task design section, we describe the association of the tasks with the three problem types as identified by Lamon and Cramer and Post and discussed how the tasks engaged students in proportional reasoning in a meaningful way.

## 4. Materials and Methods

### The Added-Sugar Activity

According to Hyman [28], reading food nutrient labels is an excellent way for an individual to become familiarized with the quality of food and make healthy dietary choices. Nutrient labels also allow mathematics educators to design creative mathematics instructions while encouraging students to use mathematics to evaluate the quality of food. Consistent with Hyman’s [28] proclamations, in this study, we designed three integrated tasks that started with students observing the pictures of four food products and their nutrient labels (Figure 1). The four food products were:a packet of Oreo cookies,a can of Coke,a 40 oz container of Chobani Greek yogurt, anda 15.3 oz box of Kellogg’s Honey Smacks breakfast cereals.

We asked the students to check their nutrient labels and note down the (i) number of servings and (ii) quantity of added sugar in each serving (in grams) in the worksheet: Count of Added Sugar (Figure 2). Next, we facilitated a brief discussion on obesity and talked to students about the different factors that cause obesity and its impact on human health. We introduced students to the daily added sugar limit of nine teaspoons for males and six teaspoons for females as imposed by the American Heart Association (AHA) and informed them that one teaspoon is equal to 4 g of sugar. We asked students to calculate the number of teaspoons of added sugar in each serving of the four food products. We identified this task to be consistent with Lamon’s [6] associated sets tasks, where the connection between teaspoons and quantity of sugar (in grams) per serving is not apparent, and it might not seem evident to the students unless we provide them with the conversion factor of 1 teaspoon = 4 g.

After students calculated the teaspoons of added sugar present in one serving of the food products, we asked them to shift their focus to the number of servings present in the entire packet of each food product and calculate the total teaspoons of added sugar in the whole package. Students used the teaspoons of added sugar per serving and the total number of servings to calculate and find the total teaspoons of added sugar in the entire packets of the food products. This task was coherent with Lamon’s [6] stretchers and shrinkers activities. A continuous one-to-one ratio preserving mapping existed between the two quantities of added sugar per serving and the number of servings. Students multiplied the two quantities to scale up the teaspoons of added sugar per serving and calculate the total teaspoons of added sugar in the entire packets of the food products.

Apart from identifying the first two tasks as associated and stretchers and shrinkers activities, we also identified them as missing value problems as defined by Cramer and Post [10]. Students would be given three pieces of information and would be asked to find the fourth one. For example, in the first task, students were asked, if one teaspoon contains 4 g of sugar, how many teaspoons will contain 17 g (for one serving of Oreo) of sugar? To solve the task, students could use the three given pieces of information, unit serving, grams of sugar/teaspoon, and grams of sugar/serving, and calculate the number of teaspoons of sugar in one serving of Oreo. The next task prompted students to calculate if one serving of Oreo contains 4.25 teaspoons of added sugar, then how many teaspoons of sugar would be present in the entire packet of Oreo (contains ten servings)? To work on this problem, students could use the three given pieces of information, unit serving, teaspoons of sugar/serving, and the total number of servings, and calculate the total teaspoons of added sugar in the entire packets of Oreo.

In the third task, we asked students to consider the daily limit of sugar recommended by American Heart Association (AHA) and find how many servings of each food product would an individual consume to reach the AHA threshold. During the design process, we considered this task to be a missing value problem, where students would use AHA’s daily limit of sugar (in teaspoons), unit serving, and the number of teaspoons of added sugar per serving to calculate the number of servings that would comprise the AHA’s daily limit. Consistent with Cramer and Post [11] and Lamon [6], we conjectured that students’ engagement with these three activities, embedded in the context of added sugar, would help us assess their reasoning around proportional quantities. In the following paragraphs, we introduce our research participants and settings and then discuss how the participants reasoned proportionally as they engaged with the tasks.

Participants and Settings

In this research, we conducted two case studies with three students for an in-depth examination of students’ interaction with the integrated tasks. All the sessions we conducted in an online setup due to the unprecedented situation caused by the novel COVID-19 virus. The three participants of the study, Lenny, Ela, and Kaya were middle school students, one of whom were from the Midwest region of the United States, and the other two were from the West coast. The ages of the students ranged between 11 and 13 years. The sessions were conducted on Zoom, and the conversations were audio and video recorded for future data analysis. The first author conducted the hour-long online sessions. The worksheet (a .dox file) and the presentation slides (a .ppt file) used in the study were emailed to the students ahead of time to avoid any potential technical difficulties during the sessions. The meetings were password protected, and the joining information was only accessible to the students and the first author.

For data analysis, the session videos were transcribed. To get an insight into students’ proportional reasoning and their conceptual understanding of proportionality, we relied on the frameworks proposed by Lamon [6] and Cramer and Post [11] and performed a two-tier exploratory data analysis. In the first tier, we investigated how students perceived the types of the tasks as they worked on them, and in the second tier, we analyzed the strategies students used to solve the different types of tasks. For example, in tier 1 analysis, if we find that a student identified a particular task as a missing value problem, then in tier 2, our goal was to analyze the strategy the student used to solve the missing value problem. In the following paragraphs, we present our findings through the three tasks and discuss how added sugar provided students with an authentic and meaningful source to engage in proportional reasoning and how the concept of proportionality provided students with an opportunity to identify the quality of the four food products.

## 5. Results

### 5.1. Task 1

In the first task, when we asked Lenny to calculate the number of teaspoons of added sugar present in one serving of Oreo, Lenny used the cross-product algorithm strategy, an extremely efficient and commonly used method [11]. He set up a proportional relationship between the given quantities and performed cross-product to calculate the answer. Lenny noted that each serving of Oreo (two cookies) contains 17 g of added sugar, which is equivalent to $\frac{17}{4}$ teaspoons of sugar. Aimed at unraveling the strategies, the first author asked Lenny to explain his solution. He said:

First, we know that one teaspoon is equal to 4 g, and we need to figure out, the 17 g is how many teaspoons… So, basically, what I did was, um, I just said one by x is equal to four over 17 (1/x = 4/17) and cross multiplied them. So, you get 17 is equal to 4x, you divide four on both sides, and you get x is equal to 17 over 4 (x = 17/4).

To capture Lenny’s reasoning, the first author further asked him to explain why he divided one by x? Lenny said, “it’s a comparison. So, on the top, we got what we were given. Um, so one teaspoon of one to four grams, but in the bottom, as a denominator, we’re trying to see, we were trying to solve for it”. Lenny also used the same cross-product algorithm strategy for calculating the number of teaspoons of added sugar present in each serving of the other food products.

Like Lenny, Ela and Kaya also determined that one serving of Oreo will contain 17/4 teaspoons of added sugar. Ela noted that one teaspoon of added sugar equals 4 g of sugar; hence, she said, “I take the total number of sugar in grams divided by four.” Like Ela, Kaya also said that “since it was 17 g of added sugar in the Oreos, I divided 17 by four.” When the first author asked Ela and Kaya to explain their approaches, the students did not explicitly explain why they divided 17 by 4, but their responses suggest that they used the cross-product algorithm [11] to convert the total quantity of sugar from grams to teaspoons.

### 5.2. Task 2

After students calculated the teaspoons of added sugar in one serving of each food product, they calculated the total teaspoons of added sugar present in the entire packets. To solve this stretcher and shrinker activity, different students used different strategies. For example, to calculate the total teaspoons of added sugar present in the entire packet of Oreo, Lenny used the three pieces of given information and used the cross-product algorithm to find the value of the fourth variable. In other words, Lenny considered this task as a missing value problem [11]. When the first author asked Lenny how many teaspoons of added sugar would be present in the entire packet of Oreo, he said, “for two (Oreo cookies) there’s 4.25 teaspoons of added sugar”, so there would be “42 teaspoons and half a teaspoon” of added sugar in the entire packet of Oreo. To better understand Lenny’s strategy, when the first author asked him to explain his solution, he said:

There are 10 servings per container. (One) serving is two cookies. So, 10 times two is 20 (cookies)…, so two over 20 is equal to 4.25 over x ($\frac{2}{20}=\frac{4.25}{x}$). Um, we could multiply it, but we could simplify it here. So, there’s one. So, it’s two over 20 ($\frac{2}{20})$. So, there’s one in 10 ($\frac{1}{10})$…And we could say that x is …the number of teaspoons in the total would be 10 times 4.25.

Lenny’s response indicates that he used the cross-product algorithm to find the solution. He used the three pieces of given information, unit serving, teaspoons of added sugar/serving, and the total number of servings, to find the fourth missing value, i.e., teaspoons of added sugar in 10 servings. Although Lenny used the cross-product algorithm to calculate the total teaspoons of added sugar present in Oreo, he used a different strategy for Chobani. He considered that each serving of Chobani Greek yogurt contains 3.5 teaspoons of added sugar, and an entire container contains four servings; hence, the total number of added sugar present in the whole container of Chobani Greek yogurt is “3.5 times four”. When the first author asked Lenny why he multiplied 3.5 with four, he said:

Um, because we found a one for one serving, and now we’re just trying to find it for four servings because that’s how many servings there are in a package. So, it’s just one serving plus one serving plus one serving plus one serving is equal to the total cup. So, another way to put that is just, um, the 3.5 times four.

To solve this stretcher and shrinker task, Lenny took an additive approach. Instead of directly multiplying 3.5 by four, Lenny added the number of teaspoons of added sugar per serving four times and thus scaled up the quantity of added sugar per serving by the total number of servings.

During their engagement with the second task, Ela and Kaya used the concept of scaling up to find out the total teaspoons of added sugar present in the entire packages of the different food products. Like Lenny, they did not use the additive approach; rather, the students multiplied the quantities of added sugar per serving by the total number of servings. For instance, when the first author asked Kaya to calculate the total teaspoons of added sugar in the entire packet of Oreo, Kaya multiplied 4.25 teaspoons of added sugar present in each serving of Oreo by ten (Figure 3). She mentioned, “There were 4.25 teaspoons of sugar in one serving, and there were 10 servings in an entire package, so you would multiply 4.25 by 10 because the package had 10 servings.”

For other food products as well, Kaya and Ela used the same strategy and determined the total number of teaspoons of sugar in the whole container of Chobani, Coke, and Honey Smacks packet. For example, to calculate the total teaspoons of sugar present in the entire container of Chobani, Kaya said, “Since there are four servings in each container (of Chobani), I multiplied 3.5 (teaspoons of added sugar in each serving of Chobani) by four. So, there are 14 teaspoons of sugar in each packet of Chobani”. Ela also used the same strategy to explain how she got the number 54 as an answer to how many teaspoons of added sugar are present in the entire package of Honey Smacks. Ela explained, “I divided the amount of sugar (in grams) in a serving by 4, which gives 4.5 as the number of teaspoons in a serving, then I took 4.5 and multiplied that by 12 since there are 12 servings.”

### 5.3. Task 3

In the third task, students were asked to calculate how many servings of each food product would an individual consume to reach the AHA threshold. For some food products, students used the concept of proportionality and calculated the number of required servings; for the others, they did not follow any strategy that could be classified under Lamon [6] or Cramer and Post’s [11] framework of proportional reasoning. For example, when we asked Lenny how many cans of coke one could drink to reach the daily sugar limit, he said, “the total teaspoons count in one Coke is 9.75. And for men it’s, um, nine teaspoons. So, you would have to drink less than one bottle, less than one Coke”. The response suggests that Lenny did not do any mathematical calculation to find the exact proportion of coke that would contain nine teaspoons of sugar. Instead, he compared the daily AHA limit of nine teaspoons with the 9.75 teaspoons of added sugar present in one can of Coke and decided that since 9.75 is greater than nine, one would have to drink less than a can of coke to reach the daily threshold.

Lenny used the concept of proportionality to find the number of servings of Oreo that would meet the AHA limit. When the first author asked Lenny, “how many Oreo cookies would you have to eat to reach the daily sugar limit as suggested by AHA?”, he said, “So, we did nine teaspoons. And each serving, which is two cookies, have 4.25 serving. So, it’s basically, if we do, um, 9 by 4.25 ($\frac{9}{4.25})$, you’ll get around 2, um, so you get two servings in which would be four cookies.” When we asked Lenny to explain his strategies, he said, “In case of Coke is 9.75. So, you would just have to drink less than a Coke. Um, like not a whole bottle, but for this one, um, there was a little bit more room, so I divided four, nine over 4.25 to see how many servings we can get.

Ela and Kaya did not use proportional reasoning to calculate the number of servings of each food product that would constitute the daily sugar limit on the third task. Instead, they compared the teaspoons of added sugar present in each serving of the food products with the AHA’s daily limit of sugar for women and estimated their answers. For instance, when the first author asked Ela and Kaya how many cans of Coke one could drink to reach the daily sugar limit, Kaya said “none”. Likewise, when we asked the students to estimate the number of servings of Oreo cookies that an individual can eat to reach the AHA’s threshold, Ela said “one”. To further explain her response, Ela said, the answer would be one “because of the 4.25”. She further added, “you can’t eat anymore”, otherwise it would cross the daily sugar limit of six teaspoons. For each of the food products, Ela and Kaya found the whole number of servings that would stay within the AHA’s limit for each of the food products. Neither of them engages in any mathematical calculations to determine the proportion of a serving that would constitute the six teaspoons. Instead, they considered the teaspoons of added sugar present in each serving and decided if the inclusion of another serving would remain within the threshold.

### 5.4. Students’ Understanding of Added Sugar through Proportional Reasoning

Apart from investigating how the issue of added sugar generates an opportunity for the learners to reason between proportional quantities, this study also intended to investigate if the mathematical concept of proportionality helps students identify the quality of food products that are part of their daily lives. To access students’ prior knowledge and awareness about food nutrients, especially added sugar, at the beginning of the sessions, the first author asked the students to list some factors that they consider important for any food purchase. In response, Lenny mentioned that he considers price and the quality of food products to be important. Although he said that to check the quality of food, he seldom checks the food-nutrient labels, he emphasized the importance of the practice. He mentioned checking food nutrient labels would help an individual to decide, “Does it (food) have any nutrient(s), when you’re giving it to your body to see if you’re getting the right nutrients.” A similar response was also made by Ela and Kaya. When they were asked to share the strategies, they use to determine the quality of a snack, Ela said that she reads the nutrient labels and checks the different nutrients, especially added sugar. When we asked her how much sugar is reasonable for an individual to consume in a day, she mentioned, “two to three grams”. Kaya’s estimation was different from Ela’s. She mentioned five grams of sugar to be a reasonable estimate for an individual. Although the students’ responses indicate their awareness of the harmful effects of added sugar on human health, they guessed the optimum quantity much lower than the AHA threshold.

In the study, as students checked the nutrient labels of the four food products and noted down their quantity of added sugar, we asked them to identify if any of the food products were healthy. In response, Lenny said that none of the products were healthy due to excess quantities of added sugar. The following excerpts capture the conversation that the interviewer had with Lenny:Interviewer: Would you consider the food products healthy? The four food products?Lenny: Um, no.Interviewer: Like none of them are healthy?Lenny: No, I was noticing, but Chobani had like 14 g of protein and it has same14 g of sugar. So, I guess I could, that could be considered a healthy abovethese three, the other three choices.Interviewer: What about Kellogg’s? It is a breakfast cereal.Lenny: Yeah. But there’s like 17 and 18 g. I mean the 18 g of sugar. It’s probably not something you should have in the morning.

Although Lenny identified that all the four food products had high quantities of added sugar, from our conversation with him, we found no evidence based on which we can claim that Lenny had a correct perception about the optimal quantity of added sugar that an individual can consume in a day.

In this study, students used proportional reasoning to measure added sugar in different food products and compare them with the AHA limit. They were surprised by their findings of added sugar. Two broad themes emerged from students’ excerpts: the quantity of added sugar that they unknowingly consume every day and the quantity of added sugar present in yogurts and breakfast cereals, which are usually considered healthy. Lenny said, “I found the Honey Smacks to be surprising because they have the second most sugar (out of the four food products); I was really surprised about the total teaspoons of sugar in the Honey Smacks.” Lenny knew that Oreo is unhealthy and contains a high quantity of added sugar; he mentioned, “Oreo is not that healthy, because our parents keep telling us”. He was surprised by the high quantity of added sugar present in breakfast cereals since they are often advertised as healthy and a good source of daily nutrients.

When the first author asked the same question to Ela and Kaya, Ela said, “(The quantity of added sugar) surprised me because how much sugar a girl can eat in a day.” Referring to the quantity of added sugar present in each serving of Oreo (two cookies), Kaya said, “I usually get like three Oreos, and then two is a lot because it’s almost how many you should have.” She further added, “Yeah, it was really surprising how much sugar there was in each of the things.” The food products that the students explored were commonly consumed by them, but from their conversation, it seemed that they were not aware of the quantity of added sugar they should ideally consume, and thus, their intake of sugar was higher than the recommended limit.

## 6. Discussion

The main purpose of this study was to investigate if a task on added sugar provides students with a meaningful context to engage in proportional reasoning and if proportional reasoning plays a pivotal role in helping students identify the quantity of added sugar present in different food products and thus decide the quality of food that they intake. As research suggests, familiar and authentic contexts enhance students’ reasoning around proportional quantities [15,16], the results of this study indicate the same. The context of added sugar served the students with a natural and meaningful scope to reason between different proportional quantities. Throughout the study, the students did not mention their prior experience with proportionality, nor did we ask if they were formally introduced to this concept at school. Although we used proportionality as a task design framework, we did not prompt students’ attention to the specific mathematical concept; instead, we wanted the students to develop their own mathematical strategies and in-sights to address the issue of added sugar. Students used proportional reasoning to solve the three tasks. They used cross-product algorithms and the strategy of scaling to solve the missing value and the stretcher and shrinker activities. Although Burgos and Godino [41] suggested that “students rely mainly exclusively on the cross-multiplication algorithm for solving proportionality tasks, disregarding that there are often better strategies to solve these problems” (p. 19), none of the participating students in this study used a single strategy for all the three tasks. Rather, we identified a relationship between the way students identified a task and their strategy to solve the task. When students identified a task as a missing value problem, they used cross-product algorithm, whereas for tasks that students identified as stretcher and shrinker activities, they used the strategy of scaling up and down. We think that the informal setting and the authentic context of added sugar might have helped students look beyond algorithms, develop their sense-making tools, and use multiple strategies relevant to given situations.

This study intended to introduce students to data on added sugar and encourage students to use mathematical reasoning to analyze the data and develop a scientific mindset and good decision-making skills to make better food choices. As the findings suggest, at the beginning of the sessions, the participants were aware of the harmful effect of added sugar; however, they did not know the threshold of the daily sugar consumption as suggested by AHA. This study familiarized the students with the daily sugar intake limit and prompted them to use proportional reasoning to calculate and compare the quality of different food products. Students were surprised by the added sugar content in yogurt and breakfast cereals, which are commonly advertised as healthy food. Dixon et al. [42] identified such a tactic of promoting unhealthy food products by highlighting their positive nutritional attributes as the nutritional content claim. For instance, advertisers often promote breakfast cereals as a good source of fibers but overlook the added sugar content of the cereals, which fails the Institute of Medicine’s nutritional recommendations for healthy food [19]. Such misinformation can put students in a dilemma about deciding the nutritional values of food products and make them more susceptible to unhealthy foods. In this study, we see such confusion in Lenny when he was unsure if 14 g of protein would make Chobani a healthy choice despite having the same quantity (14 g) of added sugar.

## 7. Conclusions

Childhood obesity is a severe problem in today’s society, and as educators, it is our responsibility to encourage our future generation to become mindful of their dietary choices and take responsibility for their health. As an initiative, in this study our team designed an integrated science and mathematics lesson and implemented it on three middle school students. We believe that more such curriculum integration is necessary to provide students a platform to understand health science through mathematical reasoning. Such curricula will not treat mathematics as a mere tool to perform scientific calculations nor use the concerned topic as a context for applying mathematical skills; instead, it would prepare the learners to explore different health-related issues through numerical evidence. It would provide the students with a mathematical language and help them develop a data-driven argument to talk about quality and access to healthy food products.

## Figures and Tables

**Figure 1 ijerph-18-12821-f001:**
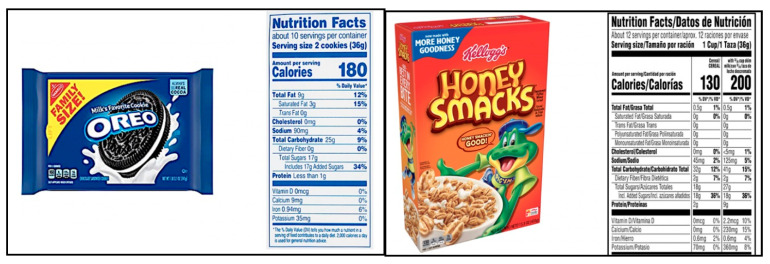
Nutritional Labels of Food Products.

**Figure 2 ijerph-18-12821-f002:**
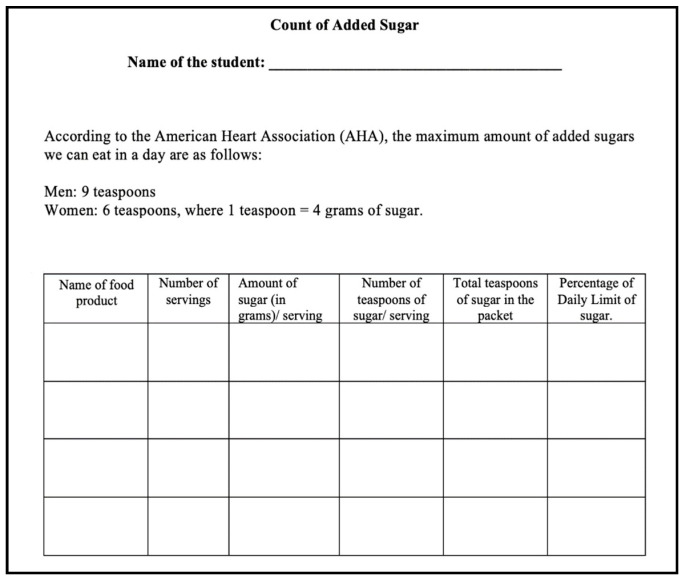
Worksheet: Count of Added Sugar.

**Figure 3 ijerph-18-12821-f003:**
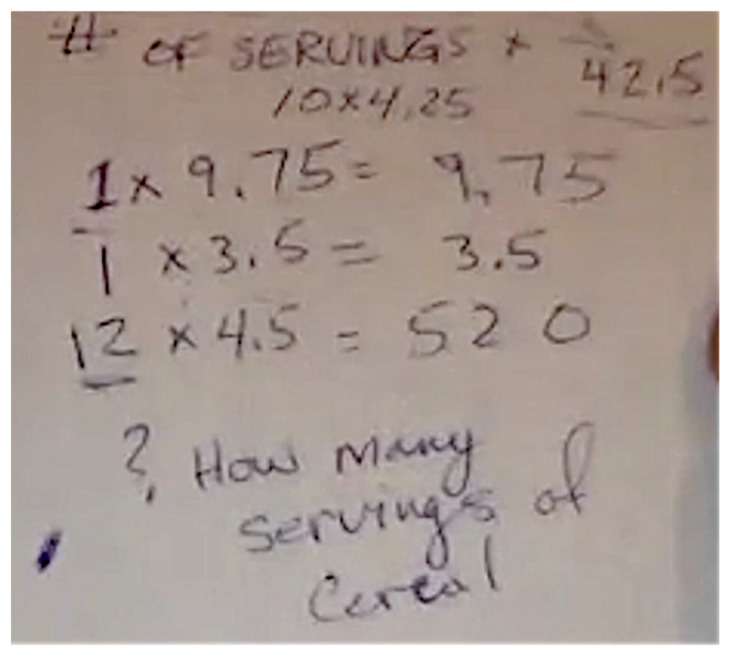
Kaya’s calculation.

## Data Availability

The data presented in this study are not publicly available because the IRB prevents us from doing so. The Cayuse IRB suggests that “the data that will be collected is considered to be private information that the individual can reasonably expect will not be made public or collected within a context that the individual would not otherwise expect to be observed or recorded”. Accordingly, in this study, we have data in two forms, video recordings and written transcription, only accessible to the PI and the Graduate Research Assistant.

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
