# Peer review of "Eating Healthy: Understanding Added Sugar through Proportional Reasoning"

_ijerph, 2021, doi:10.3390/ijerph182312821_

Round 1
Reviewer 1 Report
- I think the text is too full of many concepts should be made more concise and immediate towards the readers
- -The sample is too small. Why the authors chosed only 3 people ?
Author Response
Dear Reviewer,
Thank you again for your time and support. We appreciate the time and effort that you dedicated to providing feedback on our manuscript and are grateful for the insightful comments on and valuable improvements to our paper. We have incorporated most of the suggestions you made. Those changes are highlighted within the attached manuscript.
Best,
Debasmita and Hong

Reviewer 2 Report
Eating Healthy: Understanding Added Sugar through Proportional Reasoning
The main question addressed by the research are:
• Does a task developed on added sugar provide students with a meaningful context to engage in proportional reasoning?
• Does proportional reasoning help students identify the quantity of added sugar present in different food products and thus engage students in a discussion about food quality?
I think it is an interesting study, because the activities they propose to work on proportionality are close to the reality of the students and make them reflect on whether a meal is healthy or not, it helps to combat childhood obesity. I think it is an activity that can be extended to any school in the world.
I think that his most interesting contribution is to unify the subject of science and mathematics, it is a small activity, but it serves to open the way and motivate more colleagues.
For the methodology I think it is necessary to specify if they are based on an author to carry out the written test and interviews. I advise reading:
Understanding of Inverse Proportional Reasoning in Pre-Service Teachers
Ismael Cabero Fayos, Maria Santagueda Villanueva, Jose Vicente Villalobos-Antúnez, Ana Isabel Roig Albiol. EDUCATION SCIENCES. Num. 11. Vol. 10. pp. 1-19. 2020. Científic.
The conclusions presented answer the initial questions, are supported by the bibliography presented in the theoretical part. Finally, the bibliography is correct and up-to-date. I conclude the
report by congratulating the authors because it is a new and very interesting topic.
I make the following report and I remain at your disposal for any questions.
Author Response

(The authors gave the same response as above.)

Reviewer 3 Report
The article deals with a current topic, namely the connection between the teaching of mathematics and real life. The authors chose the right context for the tasks and the form of research. I consider the results of their research to be interesting and enriching for readers, especially for the mathematics teachers. The introduction and theoretical framework of the article are excellent. But some other parts need to be improved. My comments are:
- There is no information on the age of the respondents, so it is difficult to assess whether the tasks were adequate to the age and mathematical knowledge of the respondents. It is common in Europe for 14-year-olds (primary school) to do this.
- The results describe the procedure again; I recommend that only the findings are described in this section
- The weakest part of the article, in my opinion, is the Discussion, in which authors should incorporate their results into existing knowledge and point out the benefits of their own research. Instead, they re-describe what they wanted and what they did not want to do in the research. In addition, they describe their findings in general, for example: "We found strong evidence that exhibits how the meaningful context of added sugar enhanced students' engagement with the tasks and served the learners with the scope to engage in proportional reasoning."
What evidence did they find? Is the evidence new compared to what we already know?
I recommend rewriting this section.
Author Response

(The authors gave the same response as above.)

Round 2
Reviewer 1 Report
No comments
Reviewer 3 Report
The authors rewrote the article in a form to which I have no objections and I recommend publishing the article.